# A Rapid and Specific Real-Time PCR Assay for the Detection of Clinically Relevant Mucorales Species

**DOI:** 10.3390/ijms232315066

**Published:** 2022-12-01

**Authors:** Massimiliano Bergallo, Vivian Tullio, Janira Roana, Valeria Allizond, Narcisa Mandras, Valentina Daprà, Maddalena Dini, Sara Comini, Lorenza Cavallo, Stefano Gambarino, Anna Maria Cuffini, Giuliana Banche

**Affiliations:** 1Cytoimmunodiagnostic Laboratory, Department of Public Health and Pediatrics, University of Turin, Piazza Polonia 94, 10126 Turin, Italy; 2Bacteriology and Mycology Laboratory, Department of Public Health and Pediatrics, University of Turin, Piazza Polonia 94, 10126 Turin, Italy

**Keywords:** diagnostic microbiology, filamentous fungi, Mucorales, mucormycosis, primer and probe design, TaqMan MGB assay

## Abstract

Infections triggered by filamentous fungi placed in the order Mucorales, phylum Zygomycota, can cause serious harm to immunocompromised patients. Since there is lack of a standardized PCR (polymerase chain reaction) assay for early diagnosis of this fungal infection, this work was aimed to develop a new PCR assay able to detect the presence of Mucorales genera in clinical specimens. Here, we describe a novel diagnostic TaqMan MGB probe assay for precise and rapid detection of the most common clinical species of Mucorales. Zygomycete-specific oligonucleotides were designed to specifically amplify and bind highly conserved sequences of fungal 28S rRNA gene. Additionally, we succeeded in differentiating Mucorales species (i.e., *Rhizopus*, *Lichtheimia*, *Mucor*, and *Rhizomucor*) in artificially infected serum samples, suggesting that the quantitative capability of this real-time PCR assay could potentially optimize the diagnosis of mucormycosis.

## 1. Introduction

Mucormycosis consists of a broad range of infections triggered by filamentous fungi of the order Mucorales, phylum Zygomycota. The fungal conidia enter the human organism by ingestion, direct inoculation or inhalation. *Rhizopus arrhizus* (previously *Rhizopus oryzae*) is the most widespread species in the world. Other isolated fungi are part of the genera *Syncephalastrum*, *Cunninghamella*, *Lichtheimia*, *Rhizomucor*, *Mucor*, *Actinomucor*, *Apophysomyces*, *Cokeromyces,* and *Saksenaea* [1,2]. In the complete review by Jeong et al., *Lichtheimia* spp., *Mucor* spp., and *Rhizopus* spp. accounted for 75% of all cases [3].

The most important conditions that lead to mucormycosis include transplantation, diabetes mellitus, with ketoacidosis or not, corticosteroids, hematological malignancies and other malignancies, neonatal prematurity and malnourishment, prolonged neutropenia, illicit intravenous drug use, trauma, and iron overload [4]. One of the most important problems related to mucormycosis is that a delayed misidentification and/or diagnosis of the species involved can be fatal for the patient [5,6].

In this scenario, isolation of Mucorales from bronchoalveolar lavage fluids (BALF) or cultured tissues is often challenging [2]. In addition, urine and blood tests are often negative for mucormycosis, while biopsies may contain non-viable hyphae due to sample processing artifacts [2,6]. Indeed, viable hyphae are found in only a small number of clinical specimens, reducing the chances of obtaining fungal cultures [7]. Serological tests that are not based on cultures are currently available for diagnosis of systemic fungal infections. These serum markers, however, such as 1,3-beta-d-glucan (BDG) and galactomannan from *Aspergillus* spp., come from cell wall elements, which do not occur in Mucorales [8]. Therefore, although a positive BDG or galactomannan result may indicate fungal infection with pathogens other than mucormycosis, these tests are not capable of identifying a specific pathogen. Specific serum tests for mucormycosis are currently not available. Molecular methods, including those based on polymerase chain reaction-based approaches, are increasingly used due to their ability to enhance detection in tissues, and often help with species-level identification, through targets such as the internal transcrite spacer or ribosomal RNA 18s [9,10,11]. Other noninvasive approaches to fungal identification continue to be explored, including next-generation sequencing, gene expression profiling, and breath-based metabolomics [4]. Most of the molecular evaluations were based on pan-fungal PCR tests based on internal transcribed spacer (ITS) or multiple gender-specific PCR tests in real time to assess the most relevant Mucorales implicated in human infection, including *Lichtheimia* spp., *Rhizopus* spp., *Mucor* spp., and *Rhizomucor* spp. [12,13]. These PCR tests always need to be standardized and clinically validated.

Here, we aimed to describe a novel and reliable quantitative polymerase chain reaction (qPCR, syn. real-time PCR) protocol for rapid fungal DNA identification, which could potentially be applied to Mucorales DNA testing in the clinic routines.

## 2. Results

### 2.1. Samples Analyzed

All nine Mucorales strains used in the study resulted amplifiable by the specific qPCR developed (Figure 1). The figure shows the amplification plot of the mycelial tufts. Although we used the same inoculum of the fungi, the different efficiencies of DNA extraction and amplification of the targets reflect the observed variation in Cq. All samples used for the specificity assay show no amplification plot.

### 2.2. PCR Performance

qPCR efficiency and sensitivity were calculated according to a standard curve, as described in the Material and Methods section. In our protocol, qPCR efficiency typically ranged between a value of 3 and 4. The assay sensitivity was determined according to the lowest standard dilution measurable in replicate amplifications at 100% frequency. The PCR sensitivity was 10 copies/reaction, and the amplification was linear up to 10^2^ copies of pZIGO1 (positive plasmids control) (*Rhizopus*, *Lichtheimia*, *Mucor*) or pZIGO2 (*Rhizomucor* and *Cunninghamella*). A dilution of 10° copies/reaction of pZIGO1 and pZIGO2 was detected at a frequency between 3 and 10%—the frequencies were 10% for *Rhizopus* and *Mucor* (pZIGO1), 10% for *Rhizomucor* and *Cunninghamella* (pZIGO2), and 3% for *Lichtheimia* (pZIGO1), whereas a dilution of 10^−1^ copies/reaction at 0% frequency was observed for all plasmids tested (Figure 2).

The correlation coefficient (R^2^) was greater than 0.990, indicating excellent replicate consistency.

The coefficient of variation (CV) in the log10 concentration values was used to express the reproducibility. The CV value of the C_t_ was estimated using several concentrations between 10^2^ and 10^4^ standards within a single run (*n* = 10) or different run experiments (*n* = 10) (Table 1).

The variability was evaluated on nine Mucorales strain samples in duplicate in five independent experiments. We observed a CV median (5th, 95th) of 3.2% (0.15–13.5%) for *Rhizopus*, 2.9% (0.15–7%) for *Mucor*, 6.5% (3.3–16.7%) for *Lichtheimia*, 4.1% (0.32–11.4) for *Rhizomucor*, and 3.3% (0.22–6%) for *Cunninghamella*. Furthermore, the specificity of our PCR in the detection of Mucorales was confirmed by the observation that none of the other pathogens tested gave a false-positive result.

Next, we made serial dilutions of the pZIGO1 and pZIGO2 vectors, all comprised between 10^9^ and 10^−1^ copies/reaction, so that we could assess the dynamic range of Mucorales quantification through our PCR assay. Remarkably, we managed to quantify *Rhizopus* from 10^9^ to 10^1^ copies/reaction, with a dynamic range (DR) of 10^9^–10^1^ copies/reaction (R^2^ = 0.999), without having to load a diluted sample. Furthermore, *Mucor* could be detected from 10^9^ to 10^1^ copies/reaction, with a DR of 10^9^–10^1^ copies/reaction (R^2^ = 0.999), *Lichtheimia* from 10^9^ to 10^2^ copies/reaction, with a DR of 10^9^–10^1^ copies/reaction (R^2^ = 0.990), *Rhizomucor* from 10^9^ to 10^2^ copies/reaction, with a DR of 10^9^–10^1^ copies/reaction (R^2^ = 0.990), and *Cunninghamella* from 10^9^ to 10^2^ copies/reaction, with a DR of 10^9^–10^1^ copies/reaction (R^2^ = 0.999) (Figure 3).

### 2.3. Applicability of the qPCR Method

To assess the possible applicability of qPCR in clinical trials and to measure its sensitivity, human serum harvested from healthy donors, who gave their written informed consent, was spiked with 1 × 10^4^ conidia/mL of *Rhizopus*, *Lichtheimia*, *Mucor*, and *Rhizomucor*. The results revealed that the sensitivity of the qPCR method was 1000 conidia/mL of serum for *Lichtheimia* and 10 conidia/mL of serum for *Rhizopus*, *Mucor*, and *Rhizomucor*. In order to address the possible inhibition and to control the entire process, 100 ng of p-IC was added to the serum together with the conidia. In Figure 4, we show that all the spiked serum samples were amplifiable and the p-IC was always detectable.

## 3. Discussion

For the diagnosis of mucormycosis, there are methods that use pan-fungal primers targeting the ITS genomic region with the following sequencing of the amplified DNA, or methods of multiplex PCR using specific primers targeting a restricted number of mucoralean genera/species. Most of the molecular tests target the 18S ribosomal RNA genes, but also other targets have been investigated [14,15,16,17,18]. These include the 28S rDNA [19], region of the mtDNA *rnl* (large subunit rDNA) gene [20], the *cytochrome b* gene [21], or the Mucorales-specific *CotH* gene [22]. Bernal-Martìnez et al. studied a multiplex qPCR targeting the ITS1/ITS2 region with specific probes for *R. microsporus*, *R. oryzae*, and *Mucor* spp. [23]. Springer et al. [19] and Kasai et al. developed a specific qPCR targeting the 28S rDNA [14], and developed two independent Mucorales-specific qPCR assays, targeting two different regions of the multicopy ribosomal operon 18S and 28S, that are able to detect DNA from a broad range of clinically relevant Mucorales species. We developed a specific qPCR targeting 28S rDNA able to quantify and detect at genus levels more frequent Mucorales agents. Recently, Gade et al. provided evidence that the extended area of 28S rDNA may be a useful target for direct detection and identifying mucormycete and several other fungal pathogens from human tissue samples. This region is particularly useful for fungi where universal primers targeting ITS are unable to expand [24]. Additionally, Jillwin et al. recommended 28S rDNA as an optimal target for detection of Mucorales [25]. Though the region of 28S rDNA also has good sequence polymorphism for reliable fungal identification, the non-availability of reference sequences for the wide range of fungal species for this region limits its use. In order to make this area useful for routine identification, additional sequencing is required to evaluate the diversity of other fungi of clinical importance.

The internal tests that have been developed up to now use different primers and probes, which is why the absence of standardization makes it difficult to implement them in the clinical laboratory. Guegan et al. have validated a new pan-Mucorales qRCR commercial kit (Mucorgenius^®^, PathoNostics, Maastricht, The Netherlands) [26]. It was apparently a rapid diagnostic test with an overall sensitivity of 75% tested on serial blood specimens from subjects with culture-positive systemic mucormycosis, often preceding the final diagnosis by several days to weeks. Commercial testing (ready-to-use testing kit) was easy to use and all qPCR analyses tested in the study were superior to common methods for the detection fungal DNA.

In contrast to our developed methods, Mucorgenius^®^ is not able to conduct genus identification, and it cannot detect low fungal load. Clinical validation studies are needed for all these techniques. Therefore, here we investigated the ability of a modified qPCR assay to detect five clinically relevant Mucorales genera: *Rhizopus, Lichtheimia, Mucor, Rhizomucor*, and *Cunninghamella*. The performance of our qPCR assay using culture samples was highly satisfactory, with no cross-reactivity and 100% sensitivity. In particular, it seemed to significantly improve the quantification of microbial load, mainly due to its wide dynamic range, covering at least eight log10 copies of the nucleic acid template. Primers conjugated with minor groove binder (MGB) groups result in highly stable duplexes with single-stranded DNA targets, allowing the use of shorter probes for hybridization assays [27]. Importantly, due to their higher DNA affinity compared to standard DNA probes, these MGB-based probes are more sequence-specific, especially with respect to single base mismatches, and are therefore much more effective in differentiation. Mucorales genera have proven to be a highly trusted device for the diagnosis of systemic mucormycosis in immunocompromised patients.

In a study using three different qPCR methods (for *Mucor*/*Rhizopus*, *Rhizomucor* and *Lichtheimia*, and 18S ribosomal RNA genes) on sera from mucormycosis patients, Millon et al. demonstrated that this method is highly sensitive, has a low detection limit, and can detect infection 3–68 days before common methods [28]. We evaluated the performance of our assay using spiked sera. The results obtained in our study provide an excellent basis for future clinical investigations. The presence of PCR inhibitors (bile salts, hemoglobin, urea, heparin, EDTA, and formalin) in any specimen can affect target amplification by interfering with polymerase enzyme or other reaction ingredients. The PCR inhibition controls were assessed in the present study by using an internal control (IC) during standardization. In our study, the combination of DNA extraction and amplification protocol does not show an inhibition effect.

In terms of specificity, we did not detect cross-reactivity in our experiments, suggesting that this test could help exclude infections caused by non-Mucorales fungi, and thus avoid inappropriate antibiotic treatment [29].

Given our findings, it is likely that implementation of this method in clinical practice in the form of a commercially available kit may contribute to a rapid and accurate diagnosis of Mucorales, with a positive impact on disease management and patient outcomes [30]. In this regard, further studies are clearly needed to evaluate the usefulness of our test in detecting Mucorales in actual infected specimens from other specimen types such as BAL. The diagnosis of mucormycosis poses a challenge. Direct examination, culture, and histopathology remain indispensable tools, although molecular methods are enhancing. New molecular platforms are being evaluated and new fungal genetic targets are being studied. Methods to detect Mucorales DNA in blood have exhibited promising outcomes for rapid and early diagnosis and could be employed as screening tests in high-risk subjects, but still need to be verified in clinical trials.

In conclusion, the present study optimized qPCR protocol for detection and quantification of the Mucorales sequence in human clinical serum samples.

## 4. Materials and Methods

Nine clinical Mucorales strains, collected from three hospitals in Turin and from a veterinary hospital in Grugliasco (Turin), were analyzed: 1 *Lichtheimia corymbifera* (SL 209, from a diabetic woman’s nasal biopsy isolate); 3 *Mucor* spp (1 strain SL 555 from urine, a renal transplant recipient; 1 strain VT 507 from a cat skin lesion; 1 strain SL 96 from a human foot trauma); 3 *Rhizopus* spp (1 strain MOL 37 from a liver autoptic human sample, 2 strains MOL 96 and MOL 267 from human foot trauma); 1 *Rhizomucor* (VT 121 from a cat skin lesion); and 1 *Cunninghamella bertholletiae* (CTO 22 from a skin burn lesion). The Laboratory of Bacteriology and Mycology, Department of Public Health and Pediatrics, University of Turin, Italy provided all fungi. Strains were cultured on potato dextrose agar (Merck KGaA, Darmstadt, Germany) at 25 °C and identified by macroscopic and microscopic morphological methods. Fungi were cultured in duplicate on Sabouraud dextrose agar (SAB; Merck KGaA, Darmstadt, Germany) prior to testing and incubated for 24–72 h until hyphal growth was observed [29]. The mycelial clumps were suspended with 0.85% saline. They were then allowed to settle at room temperature for ten minutes. The supernatants were collected, diluted in PBS (phosphate buffer saline) to reach 2 × 10^4^ CFU/mL (colony forming units/mL), which was confirmed by colony counting on SAB agar in triplicate [31].

Mycelial tufts were filtered through cheesecloth and manually minced in 1.5 mL microfuge tubes with a micropestle by adding 500 μL of lysis buffer (500 mM NaCl, 400 mM TrisHCl ph 7.5, 50 mM EDTA pH 8, 1% SDS) at 60 C. The microfuge tubes were incubated for 10 min. After incubation at room temperature for 10 min, 150 μL E-lysis buffer (60 mL potassium acetate, 11.5 mL acetic acid and 28.3 high purity H_2_O) was added. Then, centrifugation was performed at 14,900× *g* for 5 min at 25 C. An amount of 700 μL of the supernatant was added to an equal volume of isopropanol, centrifuged, and washed with 70% ethanol. Ethanol was washed, centrifuged at 14,900× *g* for 5 min, dried, and suspended in 20 μL of high purity H_2_O [31].

Amplification was carried out by RT-qPCR (Real-Time-qPCR TaqMan MGB), using a 7500 real-time PCR System (Lifetech, Carlsbad, CA, USA) with 28S rDNA-specific primers, probes, and conditions previously described [32]. Briefly, for primer and probe design, the National Center for Biotechnology Information website (http://www.ncbi.nlm.nih.gov, accessed on 25 February 2020) was searched using the keywords “zygomycete” and “28S ribosomal RNA sequence” to identify available 28S ribosomal sequences in the Zygomycetes class. Data for real-time qPCR probe construction were also supplemented by sequence analysis of PCR products from culture-confirmed Mucorales isolates. The sequences were further examined using OligoPrimer analysis v.6.61 software (Molecular Biology Insights, Inc., Colorado Springs, CO, USA), Primer Express v.3.0 (Applied Biosystems, Cheshire, UK), BioEdit sequence alignment editor v. 7.0 software (Isis Pharmaceuticals, Inc., Carlsbad, CA, USA), and Sequencher v.4.0.5 software (Gene Codes, Inc., Ann Arbor, MI, USA) to identify suitable regions for primer and hybridization probes based on sequence homologies among five genera (Rhizopus, Lichtheimia, Mucor, Rhizomucor, Cunninghamella). The PCR mixture consisted of 1 × Master Mix (Platinum qPCR supermix-UDG with ROX, Lifetech, USA), 5 ul of specific primer probe mix: 0.3 uM forward and reverse primers, 0.3 uM 6-FAM-labeled probe for Rhizopus (RhizoF 5′-TCAGGTTGTTTGGGAATGCA-3′; RhizoR 5′-GGTTTCTCGCCAATATTTAGCTTT-3′; RhizoP 6FAM-CCTAAATTGGGTGGTAAAT-MGB NFQ); 0.3 uM forward, 0.6 uM reverse primers, 0.3 uM 6-FAM-labeled probe for Mucor (RhizoF 5′-TCAGGTTGTTTGGGAATGCA-3′; MucorR’-GGTCTCTCGCAAATATTTAGCTTT-3′; RhizoP6FAM-CCTAAATTGGGTGGTAAAT-MGB NFQ); 0.3 uM forward, 1.35 uM reverse primers, 0.4 uM NED-labeled probe for Lichtheimia (AbsiF 5′- GTACCGTGAGGGAAAGATGAAAA-3′; AbsiR 5′- TTCCCTCTTGGCAATTTCACATA-3′; AbsiP NED-ACTTTGAAAAGAGAGTTAAACAG-MGB NFQ); 0.6 uM forward and reverse primers, 0.5 uM NED-labeled probe for Rhizomucor (RMUF 5′- GGCTTCACAGAGGGTGACAATC-3′; RMUR 5′- GGAGCATGCATCGCAATAGA-3′; RMUP NED-CGTAGAGGGTCTTGAAAG-MGB NFQ); 0.6 uM forward and reverse primers, 0.5 uM FAM-labeled probe for Cunninghamella (CunnF 5′- GGGCGACATAGAGGGTGAAA-3′; CunnR 5′- GCCAAACGCCTAACCAAAAC-3′; CunnP FAM-CCCCGTCTTTGGCCT-MGB NFQ); 0.3 uM forward and reverse primers, 0.3 uM VIC-labeled probe for hexogen internal control IC (JellyF 5′- GCCTGGTGCAAAAATTGCTT-3′; JellyR 5′- TCGTTCATTTGTTCTTTTGTGGAA-3′; JellyP VIC-CAGCTATTGCAAACGCCATCGCAC-TAMRA); and 5 ul of DNA extracted samples at a final volume of 20 ul. The reactions of each individual target were run separately.

The amplification program was 1 cycle for 10 min at 95 °C, 45 cycles for 10 s at 95 °C, and for 60 s at 60 °C. Threshold cycle (Ct) values, which correspond to the number of cycles required to generate a fluorescence signal above background levels, were directly proportional to the initial log concentration of the target DNA. Three wells were loaded for each target and the resulting arithmetic mean was used as the result.

qPCR sensitivity was determined by analyzing the lowest target concentration with a 100% frequency. Ct values included within the dynamic range but outside the measurements were not included. Thus, a Ct with a value of >40 was deemed negative. Results were confirmed by direct DNA sequencing after amplification with the same primer used in qPCR. Sequence PCR was performed using the Dye Terminator Cycle Sequencing Kit 1.1 (Lifetech). The nucleotide sequences were determined on an ABI 310 automated sequencer (Lifetech).

The positive plasmid controls pZIGO1 (*Lichtheimia, Mucor*, and *Rhizopus*), pZIGO2 (*Cunninghamella, Rhizomucor*), and pIC (zebrafish) were all from TwinHelix (Rho, Italy). The plasmid was in-silico designed and contains the entire sequence amplified by specific qPCR cloned into the pUC57 vector. As a negative control (NTC), ddH_2_O was used. qPCR products were quantified by plotting values on a standard curve obtained by serial 10-fold dilutions of the pZIGO1 and pZIGO2 vectors to cover a 3-log dynamic range.

qPCR analytical variability was determined by intra/inter-assay coefficient of variation (CV). pZIGO1 and pZIGO2 standard plasmids (range = 10^4^–10^2^ copies/reaction) within a single run (*n* = 10) or different run experiments (*n* = 10) were amplified. Moreover, the amplification efficiency (E), typically ranging from 1.7 to 2.2, was calculated as 10(−1/slope). PCR sensitivity was assessed by testing DNA isolated from clinical specimens, as aforementioned, and DNA from nine Mucorales stock strains.

Further, the qPCR analytical specificity was determined by testing 2 *Malassezia furfur* (animal skin lesion), 1 *Scedosporium apiospermum* (SL 87, human ear lesion), 1 *Scopulariopsis brevicaulis* (MOL 108, human onychomycosis), 1 *Fusarium* spp. (SL 211, skin lesion), 1 *F. solani* (SL 11-8, human leg lesions), 1 *Cryptococcus neoformans* (MOL 21K, hemoculture from a immunocompetent patient), and 1 *Saprochaete capitata* (formerly *Geotrichum capitatum*, SL 199, human feces). *M. furfur* was identified and cultured according to Gupta et al. [33], while *S. capitata* and *C. neoformans* were identified based on morphological characteristics and by ID32C^®^ (BioMérieux, Rome, Italy) [34,35] and incubated on SAB agar for 24–48 h.

Additionally, we tested 50 ng of *Streptococcus pneumoniae* ATCC 6301, 1 *Staphylococcus aureus*, 1 *Legionella pneumophila* ATCC 33152, 1 *Chlamydia pneumoniae* ATCC 53592, 1 *Mycoplasma pneumoniae* ATCC 15377, 1 *Escherichia coli* (OIRM 2, from urine), and 1 *Pseudomonas aeruginosa* (MOL 10, from positive blood culture).

To evaluate the potential applicability of the PCR method and to examine its sensitivity in clinical trials, human serum harvested from healthy donors, who gave their written informed consent, was spiked with 1 × 10^4^ conidia/mL of *Rhizopus*, *Lichtheimia*, *Mucor*, and *Rhizomucor* to obtain artificially infected sera and 100 ng of p-IC. The p-IC served as the entire process control.

## Figures and Tables

**Figure 1 ijms-23-15066-f001:**
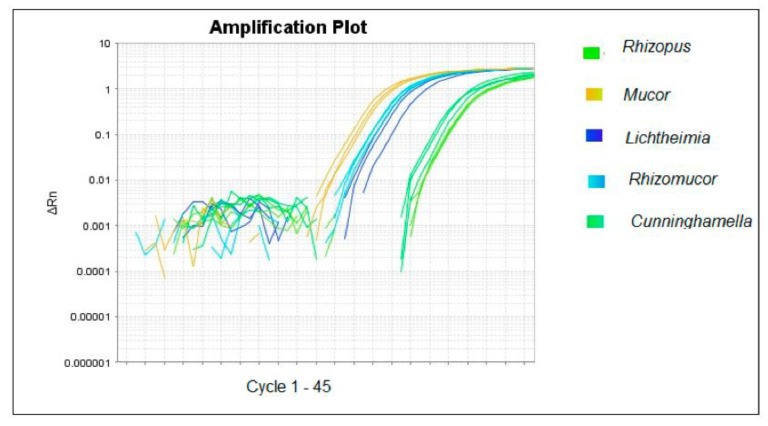
Amplification plot of Mucorales qPCR assays. The figure shows the triplicates of the analyzed samples. Green: *Rhizopus;* yellow: *Mucor;* blue: *Lichtheimia;* light blue: *Rhizomucor;* and light green: *Cunninghamella*.

**Figure 2 ijms-23-15066-f002:**
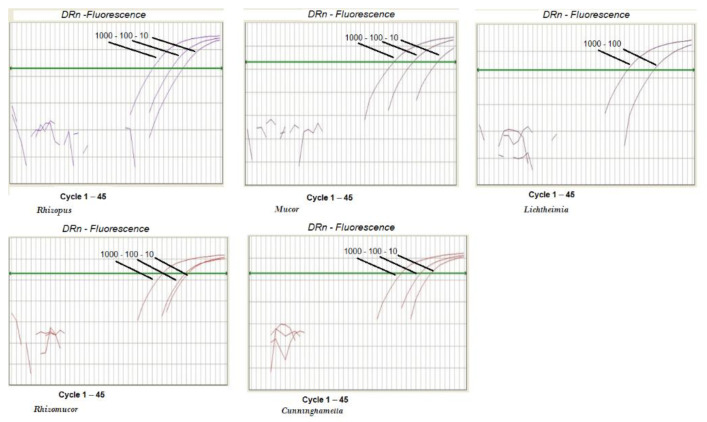
Sensitivity of five different qPCR assays for detection of Mucorales. From left to right: *Rhizopus; Mucor, Lichtheimia, Rhizomucor,* and *Cunninghamella*.

**Figure 3 ijms-23-15066-f003:**
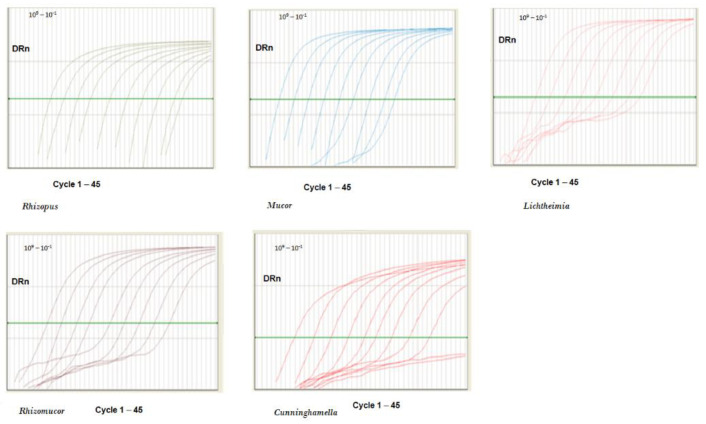
Dynamic range of Mucorales quantification by qPCR. From left to right: *Rhizopus, Mucor, Lichtheimia, Rhizomucor,* and *Cunninghamella*.

**Figure 4 ijms-23-15066-f004:**
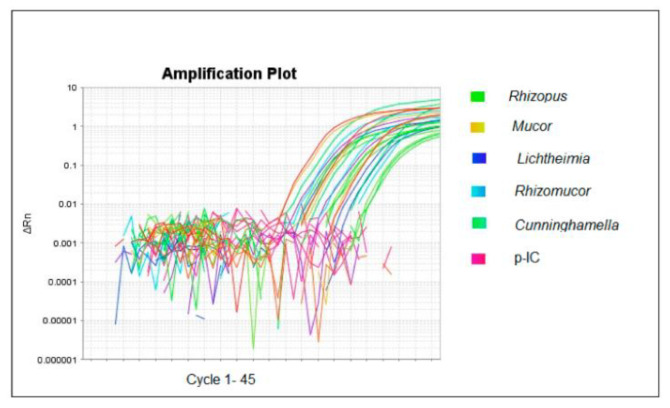
Inhibitory test of Mucorales qPCR. The figure shows the amplification plot of the analyzed samples. Green: *Rhizopus;* yellow: *Mucor;* blue: *Lichtheimia;* light blue: *Rhizomucor;* light green: *Cunninghamella;* violet: *p-IC*.

**Table 1 ijms-23-15066-t001:** Values of intra-assay and inter-assay variability expressed as a coefficient of variation (CV) for each of the indicated standard plasmid concentrations.

Standard Plasmid DNA	Intra-Assay Variability (%)	Inter-Assay Variability (%)
10^2^	0.473	0.785
10^3^	0.671	1.699
10^4^	0.633	1.345
10^5^	0.088	0.888

## Data Availability

The data presented in this study are available on request from the corresponding author. The data are not publicly available due to policy of the Department of Public Health and Pediatrics.

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
