# Peer review of "A Rapid and Specific Real-Time PCR Assay for the Detection of Clinically Relevant Mucorales Species"

_ijms, 2022, doi:10.3390/ijms232315066_

Round 1
Reviewer 1 Report
This study aims to develop a new PCR assay able to detect the presence of Mucorales genera in clinical 18 specimens, Please clarify the following main aspects:
1. Line 22: do you mean 28 SrRNA
2. Line 60: what is meant by STIs? Please write the abbreviation means along the manuscript.
3. Figure1(line 82): lef must be change into left.
4. Figure 2(line 110): lef must be change into left.
5. All the bacterial & fungal genera and scientific names must appear in italics
6. Moderate English changes required
7. The bibliographic sources cited are generally old. Introduction must be improved and authors can included these recent
18S rRNA gene sequencing for environmental aflatoxigenic fungi and risk of hepatic carcinoma among exposed workers. Journal of environmental science and health part A. 57(3), 174–182.
Impact of mannose-binding lectin gene polymorphism on lung functions among workers exposed to airborne Aspergillus in a wastewater treatment plant in Egypt. Environmental Science and Pollution Research. 29(42):63193-63201, https://doi.org/10.1007/s11356-022-20234.
Author Response
Line 22: do you mean 28 SrRNA
No, we intended the subunit 28 of the ribosome, 28S rRNA
Line 60: what is meant by STIs? Please write the abbreviation means along the manuscript.
We thank the Reviewer for the comment. STIs was changed in “ITS” and we proceeded to explain
Figure1(line 82): lef must be change into left.
Please see the revised version: “lef” was changed in “left”.
Figure 2(line 110): lef must be change into left.
Done
All the bacterial & fungal genera and scientific names must appear in italics
Done
Moderate English changes required
Done
The bibliographic sources cited are generally old. Introduction must be improved and authors can included these recent
18S rRNA gene sequencing for environmental aflatoxigenic fungi and risk of hepatic carcinoma among exposed workers. Journal of environmental science and health part A. 57(3), 174–182.
Impact of mannose-binding lectin gene polymorphism on lung functions among workers exposed to airborne Aspergillus in a wastewater treatment plant in Egypt. Environmental Science and Pollution Research. 29(42):63193-63201, https://doi.org/10.1007/s11356-022-20234.
We thank the Reviewer for the suggestion, these references are very interesting. Suggested articles were added to the references.
Reviewer 2 Report
Bergallo and colleagues submitted a manuscript entitled “A rapid and highly specific real-time qPCR TaqMan MGB method for the detection of clinically relevant Mucorales species” for publication as Communication in MDPIs Molecular Sciences.
As mentioned in the title, the aim is to present a diagnostic assay for the rapid and “highly specific” determination of pathogenic species from this order. This diagnostic TaqMan MGB probe assay should enable identification of most common clinical species of Mucorales. To achieve this goal, they use a previously published 28S approach.
Well, there is no doubt that such an assay is of great interest to the scientific community, clinical diagnostics and the readers of this journal.
Surprisingly, the manuscript is not well written. The material and methods section contains fundamental flaws that prevent a repetition of the present study. The chapter on results does not allow for a comprehension of the study results in the given form. Only three fragments are presented in which the results are described in overview and there is no reference to the large number of samples in the M+M section. Furthermore, it remains unclear why this assay should be specific.
Unfortunately, this manuscript does not meet the requirements for publication.
General comments:
1. Well, to my understanding a TaqMan™ MGB probe assay was performed. The multiplication and mixing of the terms real-time, TaqMan and qPCR should be avoided in the entire manuscript (title included) and please limit RT-qPCR for reverse-transcription qPCR.
2. The introduction is brief and focused, this is in accordance with the communication format of the manuscript.
3. The Material and Methods section has serious problems in experimental design or, at least, information gaps, resulting in heavy problems for the entire study:
3.1 The authors do not give any information on an exact naming of the examined lines and partly of the species (L215-6; this problem comes up later again, L257ff and again L280). A repetition of the experiments is thus not possible. Furthermore, no information was provided on the original assignment. This should be provided, if the department does not act as an official supplier.
3.2 If the samples originated from patients, how do handle the ethic statement?
3.3 No information on the templates in terms concentration or quality is provided.
3.4 Primer, probes and conditions were linked to Gupta et al. 2000 (reference 32), which seems to be the wrong reference. Anyhow, this part is critical for the understanding of the whole study and details have to presented here.
3.5 The information on positive controls for plasmids is insufficient with respect to the insert origin.
3.6 Information on the NTCs could not be found.
3.7 Probably overlooked by me, but where are the necessary triplicates mentioned in the TaqMan assay?
3.8 How were non-specific amplicons excluded?! No experimental evidence or in silico analyses are presented.
3.9 No target verification of the small amplicons by Sanger sequencing is provided (examples would be enough).
4. The section results
4.1 Results should start with a general overview of the sample results (unambiguously assigned and ordered). The present results section lacks such an overview and also detailed information on all samples mentioned in the M+M section.
4.2 Section 2.1 make the claim “Being less than 10up2 copies, the quantification of Mucorales was found to be highly reproducible.” but Figure 1 shows no repetition. Furthermore, the results refer to the plasmid concentrations, which are not stated.
4.3 Calculation of intra- and Inter-assay variability is not mentioned in the M+M section.
4.4 Section 2.2 describes variability and is linked to figure 2. However, no explanation is given for results of Cunninghamella. And in addition, why are no duplicates in the curves visible?
4.5 Section 2.3, too cryptic.
Author Response
ANSWER TO REFEREE 2 COMMENTS
-
Well, to my understanding a TaqMan™ MGB probe assay was The multiplication and mixing of the terms real-time, TaqMan and qPCR should be avoided in the entire manuscript (title included) and please limit RT-qPCR for reverse-transcription qPCR.
We agree with these suggestion. In the revised form (title included) we modified
-
The introduction is brief and focused, this is in accordance with the communication format of the
-
The Material and Methods section has serious problems in experimental design or, at least, information gaps, resulting in heavy problems for the entire study:
We thank the Reviewer for the precious suggestion in order to improve the manuscript. The Material and Methods section was improved.
-
The authors do not give any information on an exact naming of the examined lines and partly of the species (L215-6; this problem comes up later again, L257ff and again L280). A repetition of the experiments is thus not possible. Furthermore, no information was provided on the original This should be provided, if the department does not act as an official supplier.
We agree. Please see at pages 6-8 in the revised manuscript: we have added species information, as suggested.
-
If the samples originated from patients, how do handle the ethic statement?
The samples were from the collection and we have informed consent from patients
-
No information on the templates in terms concentration or quality is
We thank the reviewer for the comment. Please see in the revised manuscript form.
-
Primer, probes and conditions were linked to Gupta et al. 2000 (reference 32), which seems to be the wrong reference. Anyhow, this part is critical for the understanding of the whole study and details have to presented
Again, we thank the Reviewer for the suggestion. We correct the reference and add details.
-
The information on positive controls for plasmids is insufficient with respect to the insert
We thank the Reviewer for the comment. We insert in the text specific.
-
Information on the NTCs could not be
We add NTC controls in the text.
-
Probably overlooked by me, but where are the necessary triplicates mentioned in the TaqMan assay?
We thank the reviewer. We specify it in material and methods section.
-
How were non-specific amplicons excluded?! No experimental evidence or in silico analyses are
We correct and add details. Please see in the revised form
-
No target verification of the small amplicons by Sanger sequencing is provided (examples would be enough).
We insert notice in the text.
-
The section results
-
Results should start with a general overview of the sample results (unambiguously assigned and ordered). The present results section lacks such an overview and also detailed information on all samples mentioned in the M+M
-
We add results of the samples mentioned in the Material and Methods section.
-
Section 1 make the claim “Being less than 10up2 copies, the quantification of Mucorales was found to be highly reproducible.” but Figure 1 shows no repetition. Furthermore, the results refer to the plasmid concentrations, which are not stated.
The sentence was removed. It is an error.
-
Calculation of intra- and Inter-assay variability is not mentioned in the M+M section. The reference to intra and inter test variability is reported in the Material and Methods section.
-
Section 2 describes variability and is linked to figure 2. However, no explanation is given for results of Cunninghamella. And in addition, why are no duplicates in the curves visible?
We thank the Reviewer for the point. The figure 2 became figure 3, report the dynamic range of the different RT-qPCR assays and no duplicated is necessary. The result of CV about Cunninghamella are already reported in the sentence.
-
Section 3, too cryptic.
Again, we thank the Reviewer for the suggestion. We modified the section.
Round 2
Reviewer 2 Report
Bergallo and colleagues have submitted a revised version of their manuscript. The new version has already improved, but still shows too many shortcomings for publication.
Please perform the following changes:
Title:
This title cannot be used due to the following reasons:
- Remove highly, I guess that this aims for the comparison to a commercial kit without using a comparable sample set?
- Real-time RT-qPCR is a repetition in content. Change to:
“A rapid and specific Real-time PCR assay for the detection of clinically relevant Mucorales species”
OR
“A rapid and specific qPCR assay for the detection of clinically relevant Mucorales species”
Abstract:
L18, content repetition (real-time PCR or qPCR!), change “TaqMan minor groove binder (MGB) real-time quantitative PCR (RT-qPCR) assay” TO “diagnostic TaqMan MGB probe assay”
L23 (see above), content repetition, change “quantitative capability of this Real Time qPCR assay” TO “quantitative capability of this Real Time PCR assay”
Keywords:
L27 (see above), “RT-qPCR TaqMan MGB” TO “qPCR” OR(!) “TaqMan MGB assay”
Introduction
L48, Galactomannan TO galactomannan
L59-62, remove or correct at least typos …, in the spacer DNA … precursor transcript
L59, change “Internal Transcribed Spacer” TO “internal transcribed spacer”
L66, “real-time quantitative PCR (Real Time-qPCR, polymerase chain reaction)” TO “quantitative polymerase chain reaction (qPCR, syn. real time PCR)”
From L66 to L361!, apply the term “qPCR”, do not use RT-qPCR, RT-qPCR, Real Time-qPCR TaqMan MGB or whatever.
Results
Section 2.1, no explanation is given for the apparent differences in Cq values.
Discussion
L150-161, discard this part with respect to a missing statement and journal level
M+M
L277, “The PCR mixture”, please clarify in the text “multiplex assay”or “reaction mix”
L279-296, correct the formatting
L279ff, MGB NFQ or which quencher was used? Provide information!
M+M section, with respect to the result section
L331-343, additionally we tested …
- Provide template concentration for each sample or remove the sample from the study.
- S. aureus no strain or origin is assigned so far
- Explain the reason for the experiments with RNA viruses in a DNA-template based assay or remove this part!
- Provide correct/traceable ATCC numbers or remove samples!
- Provide detailed results on all listed samples including the Ct-values in the result section or remove the complete paragraph!
- Provide information on the template concentration corresponding to the results of figure 1.
References:
L384, in italics Aspergillus
Author Response
ANSWER TO REVIEWER 2 COMMENTS
-
Title:
This title cannot be used due to the following reasons:
-
Remove highly, I guess that this aims for the comparison to a commercial kit without using a comparable sample set?
-
Real-time RT-qPCR is a repetition in content. Change to:
“A rapid and specific Real-time PCR assay for the detection of clinically relevant Mucorales species”
OR
“A rapid and specific qPCR assay for the detection of clinically relevant Mucorales species” Thanks to the reviewer. Following your suggestion, in the revised form we have modified the title.
-
Abstract:
L18, content repetition (real-time PCR or qPCR!), change “TaqMan minor groove binder (MGB) real-time quantitative PCR (RT-qPCR) assay” TO “diagnostic TaqMan MGB probe assay” Done
L23 (see above), content repetition, change “quantitative capability of this Real Time qPCR assay” TO “quantitative capability of this Real Time PCR assay”
Done. Please see lines 22-23
-
Keywords:
L27 (see above), “RT-qPCR TaqMan MGB” TO “qPCR” OR(!) “TaqMan MGB assay”
Done
-
Introduction
L48, Galactomannan TO galactomannan
Please see the revised version: “Galactomannan” was changed in “galactomannan ”.
L59-62, remove or correct at least typos …, in the spacer DNA … precursor transcript
Thank you for the suggestion. We have removed the sentence. Please see page 2, line 58
L59, change “Internal Transcribed Spacer” TO “internal transcribed spacer”
Done
L66, “real-time quantitative PCR (Real Time-qPCR, polymerase chain reaction)” TO “quantitative polymerase chain reaction (qPCR, syn. real time PCR)”
Done
From L66 to L361!, apply the term “qPCR”, do not use RT-qPCR, RT-qPCR, Real Time-qPCR TaqMan MGB or whatever.
Done. We have apply the term qPCR from where suggested. Please see from page 2
-
Results
Section 2.1, no explanation is given for the apparent differences in Cq values.
We thank the Reviewer for the suggestion. In the revised form, we have explained the difference in Cq values. Please see pag 2, lines 67-73
-
Discussion
L150-161, discard this part with respect to a missing statement and journal level
As suggested by the reviewer, we deleted these parts
-
M+M
L277, “The PCR mixture”, please clarify in the text “multiplex assay”or “reaction mix”
Please see the revised version. We have clarified
L279-296, correct the formatting
We are sorry but it is not possible to improve the formatting
L279ff, MGB NFQ or which quencher was used? Provide information!
Done
M+M section, with respect to the result section L331-343, additionally we tested …
-
Provide template concentration for each sample or remove the sample from the
-
aureus no strain or origin is assigned so far
-
Explain the reason for the experiments with RNA viruses in a DNA-template based assay or remove this part!
-
Provide correct/traceable ATCC numbers or remove samples!
We deleted this part
-
Provide detailed results on all listed samples including the Ct-values in the result section or remove the complete paragraph!
-
Provide information on the template concentration corresponding to the results of figure
In the revised form, we have added the requested information. We thank the Reviewer for the suggestion.
References:
L384, in italics Aspergillus
Done
Round 3
Reviewer 2 Report
L70 : Correct phrase, Although …
L109, 2.4: Correct paragraph title
L143, Correct upper letters in the legend
L150, rnl, provide full name
L151, Bernal- …. Check phrase, it lacks a word
L172, correct typo
L173, Check phrase: Commercial testing was easy to use and all qPCR ana…; correct typo, clarify what’s the meaning of “commercial” in this context!
L176, correct typo
L238 a.o., ml or mL
L277 and L282, check spelling but more important, change phrase and clarify: “Each reaction were run in singleplex in a separate tube.” AND in L282 “Each sample was run in triplicate and the median value were used.”
L294, ddH2O, check format
L301, check spelling
Author Response
L70 : Correct phrase, Although …
Thank you for the suggestion. We have correct phrase. Please see line 70 in the revised manuscript.
L109, 2.4: Correct paragraph title
Thank you for the suggestion. In the revised form we have modified paragraph title and united paragraph 2.2, 2.3 and 2.4 in one paragraph “2.2 PCR performance”.
L143, Correct upper letters in the legend
Done
L150, rnl, provide full name
Done
L151, Bernal- …. Check phrase, it lacks a word
Done. Thank you for the suggestion
L172, correct typo
Done
L173, Check phrase: Commercial testing was easy to use and all qPCR ana…; correct typo, clarify what’s the meaning of “commercial” in this context!
Commercial in this context means "ready-to-use kit" without the need to assemble different reagents. In the revised form we have better clarified as suggested.
L176, correct typo
Done
L238 a.o., ml or mL
Please see the revised version: “ml” was changed in “mL ”.
L277 and L282, check spelling but more important, change phrase and clarify: “Each reaction were run in singleplex in a separate tube.” AND in L282 “Each sample was run in triplicate and the median value were used.”
Thank you for the suggestion. Following your suggestion, in the revised form we have modified the sentences.
L294, ddH2O, check format
Done
L301, check spelling
Done